# Role of Free-Ranging Synanthropic Egyptian Geese (*Alopochen aegyptiaca*) as Natural Host Reservoirs for *Salmonella* spp. in Germany

**DOI:** 10.3390/ani13213403

**Published:** 2023-11-02

**Authors:** Ella F. Fischer, Romy Müller, Matthias Todte, Anja Taubert, Carlos Hermosilla

**Affiliations:** 1Institute of Parasitology, Biomedical Research Center Seltersberg (BFS), Justus Liebig University Giessen, 35392 Giessen, Germany; anja.taubert@vetmed.uni-giessen.de (A.T.); carlos.r.hermosilla@vetmed.uni-giessen.de (C.H.); 2Avicare+, 06366 Köthen, Germany

**Keywords:** *Salmonella* spp., *Alopochen aegyptiaca*, alien species, anthropozoonosis, One Health

## Abstract

**Simple Summary:**

*Salmonella* is worldwide one of the most common and pathogenic bacteria causing severe gastroenteritis in humans and animals. As such, birds are natural carriers of different zoonotic-relevant *Salmonella* serovars. Consistently, Anseriformes transmit *Salmonella* spp. to humans and manifest clinical Salmonellosis. The Egyptian goose (EG; *Alopochen aegyptiaca*) represents a fast-spreading alien species in Europe, North America and Asia. This alien species prefers urban habitats such as parks, urban ponds, public swimming pools, riversides or golf courses, thereby having frequent contact with humans, wildlife and domestic pets. Increased environmental EG faecal contamination in cities brings up the question of potential anthropozoonotic pathogen spill-overs to humans, including *Salmonella*. To the best of our knowledge, this is the first study to investigate the role of the EG as a natural host reservoir of *Salmonella* but also to discuss transmission routes of salmonellosis to humans in chlorinated public swimming pools.

**Abstract:**

*Salmonella* is one of the most common and pathogenic bacteria worldwide, causing severe enteritis in humans and representing a relevant intestinal illness in One Health for young, old and immunosuppressed patients. Various *Salmonella* serovars have been described to be responsible for human Salmonellosis. Birds represent natural carriers of different zoonotic-relevant *Salmonella* serovars and Anseriformes can not only transmit *Salmonella* spp. to humans but also manifest clinical Salmonellosis. In this study, 138 scat samples (*n* = 138) of free-ranging Egyptian geese (EG; *Alopochen aegyptiaca*) were collected in Germany, including 83 scat samples from city parks, 30 samples from 14 public swimming pools and 25 fresh caecal samples of dead EG. Collected EG scat samples were examined for the presence of *Salmonella* spp. according either to the ISO 6579 (2017) norm or to a combination of bacterial pre-enrichment and specific PCR for detection of *Salmonella* DNA. All 138 analysed EG faecal samples resulted *Salmonella*-negative. Furthermore, the survival of *Salmonella enterica* subsp. *enterica* Serovar Anatum in spiked EG droppings was tested in four different concentrations of chlorinated pool water. In vitro testing demonstrated that *S*. Anatum-spiked EG droppings were still infectious for up to six hours in chlorinated pool water according to current German regulations for public swimming pools. This study is to be considered as a baseline investigation to clarify the role of synanthropic EG as natural carriers of zoonotic *Salmonella* in cities; nonetheless, large-scale epidemiological studies, including higher numbers of samples as well as more urban locations, are needed for final conclusions on the occurrence of this intestinal bacteria in neozootic EG.

## 1. Introduction

The Egyptian goose (EG; *Alopochen aegyptiaca*), colloquially known in Germany as the ‘Nile goose’, represents a fast-spreading alien species in western Europe [1,2,3]. This shelgoose species (subfamily Tandorninae) was initially brought in the 17th century as ornamental birds to parks allocated in England and in the Netherlands [4,5] (Figure 1a,c). Today they are well-established as a neozootic species throughout Germany, but with larger populations inhabiting cities in the western parts of the country. Adults can reach up to 70 cm of height [6], with an average body weight (BW) of 2.6 kg for males and 2.2 kg for females [2]. Neozootic EG prefer habitats around small freshwater bodies in combination with open grasslands [7]. Interestingly, these alien birds cover greater distances on foot than native mallards (*Anas platyrhynchos*) and mute swans (*Cygnus olor*), thereby contaminating more lawn and walkway surfaces with their droppings. Typical EG habitats are frequently found in urban parks, ponds and public swimming pools within numerous European cities [8]. As a synanthropic species, EG are well adapted to humans, therefore having short escape distances when compared to other endemic waterfowls. Consequently, many urban EG show tame behaviour toward humans and/or domestic pets, facilitating species–species interactions.

Nowadays, wild EG populations can be found with high densities in numerous German cities, parks, ponds and even in public swimming pools, resulting in serious faecal (dropping) contamination of environments, as recently reported [2,9]. Especially EG droppings on lawns, around natural water bodies or within waters may result in the eutrophication of ponds, thereby having a direct detrimental impact on freshwater ecosystems. Environmental EG dropping contamination is often underestimated, as a single adult can defecate daily approximately 0.7 kg of faeces. In addition, urban EG frequently form clusters, resulting in serious dropping contamination of popular public recreation areas for citizens and pets [10]. Consequently, many cities in western Germany have decided to engage professional hunters to limit numbers of urban EG in order to hamper pollution [2,11,12].

Despite these facts, there are only two reports in the literature focusing on the occurrence of intestinal zoonotic pathogens in synanthropic EG in Germany. One study focused on gastrointestinal parasites, identifying adults of *Echinostoma revolutum* in the gut of deceased animals [2], and another investigated bacteria and virus infections, where *Mycoplasma* spp. and influenza A were found [10]. Concerning *Salmonella* spp. infections in EG populations, there are neither reports on prevalence nor incidence, although wild EG meet criteria to act as natural hosts/carriers of zoonotic *Salmonella enterica* subsp. *enterica,* as described by Smith et al. [13]. Further, Smith et al. show in their review that Anseriformes are slightly under-represented in studies of *Salmonella* in wildlife compared to other avian orders.

Accordingly, geese and ducks (Anseriformes) are capable of transmitting and becoming ill with enteric *S. enterica* subsp. *enterica*. Therefore, not only sick EG but also asymptomatic animals may shed *S. enterica* subsp. *enterica* serovars into urban habitats [14]. The determined prevalences are strongly variable depending on the host species and the place of sampling (Table 1) [15].

Furthermore, bacteriological studies on waterfowl droppings in different urban parks of London and Yorkshire, United Kingdom (UK), showed divergent prevalences of *S. enterica* subsp. *enterica*. Detected *Salmonella* prevalences in the UK varied from 0% up to 20% in a single York park [24], whereas others showed lower prevalences of 4–5% but were still considered higher than the ones reported in Germany [23].

In contrast to waterfowls, much higher *Salmonella* spp. prevalences have been reported in wild (44.1% infected birds; *n* = 34) [20] and domestic Galliformes (1.3% infected layer flocks in 2022 in Germany; *n* = 7009) [25].

For epidemiological or nosological reasons, salmonellosis may be subdivided into systemic typhoid, para (non)-typhoid illness and acute gastroenteritis. More than 2.400 *Salmonella* serovars of *Salmonella enterica* subsp. *enterica* are known at present [26,27]. Irrespective of taxonomy, approximately 500 *S. enterica enterica* serovars are considered of zoonotic potential at present. Human infection is always initiated by the oral route, either directly by consumption of *Salmonella*-contaminated food or indirectly by contact with contaminated foodstuffs (e. g. meat, eggs, milk, cheese, mayonnaise, ice cream, weed and salads) [28]. Healthy adults might become infected by a single dose of approximately 1 × 10^4^–10^6^ bacteria, whereas neonates, young, old, as well as immunosuppressed patients might become infected with only 1 × 10^2^ bacteria [29]. The importance of human salmonellosis in Germany is clearly evidenced by the high number of non-typhoid salmonellosis cases (*n* = 13,700) for the year 2019 reported to the Robert Koch Institute (RKI) in Berlin. Thus, human salmonellosis is still considered as the second-leading cause of acute gastroenteritis in this country, resulting in a negative impact not only for individuals but also for public health issues [23]. The highest incidences are primarily reported in children under three months of age [28]. Even more devastating, 0.1% of these infant cases have lethal outcomes [29]. Particularly in the toddler cohort (12–36 months) with marked geophagy and rather low hygiene awareness, the feco-oral route and smear infections are considered epidemiologically relevant as reported for other infant-associated enteric parasites (i.e., *Cryptosporidium parvum*, *Giardia intestinalis*) [30].

Epidemiological connections between various host species and the survival of *Salmonella* in the environment are potentially responsible for complicated human salmonellosis outbreaks [31]. Consequently, *Salmonella* has the capacity to survive and multiply in waterfowl faeces, thereby remaining infectious for up to a month in contaminated environments [13,24]. Conversely, *Campylobacter* replication, detection or persistence did not occur after two days in geese droppings [32]. Thus, humans might be exposed to *Salmonella-* containing EG droppings in different urban environments, including public swimming pools.

In order to clarify the potential role of synanthropic EG as a natural wild host of *S. enterica* subsp. *enterica* in cities, here we assessed the presence of this bacteria in collected droppings in different urban environments of Germany and Luxembourg. Furthermore, to assess bacterial persistence in EG droppings as close as possible to the real scenario observed in contaminated swimming pools, in vitro studies were conducted on the survival of *Salmonella* Anatum in spiked EG droppings under different chlorinated water conditions.

## 2. Materials and Methods

The samples were taken from February 2020 to June 2021 over all seasons to avoid seasonal distortions of results, regardless of the fact that previous studies have shown no seasonal fluctuations of *Salmonella* spp. prevalences in various waterfowl species [33].

Due to the fact that EG are in general non-migratory and limited to short-distance movements [2,8], the samples were taken in five different federal states of Germany and two cantons of Luxembourg (Figure 2).

All samples were examined in a specialised *Salmonella*-diagnostic laboratory, accredited to the current DIN ISO 17025 (2018) norm for laboratories [34].

### 2.1. Scat Samples

From January 2020 to June 2021, a total of 179 synanthropic EG (*n* = 179) were searched and thereafter observed in their urban habitats. These wild EG were living either as single individuals or in small or even large groups (especially in autumn and early winter months) with up to 70 animals. Whenever spontaneous EG defecation occurred, fresh scat samples were immediately collected in sterile 10 mL plastic tubes (Kruuse, Denmark). EG in groups were regarded as a cluster and not every animal was sampled, but a representative amount of scat samples was collected, as reported elsewhere [35,36]. *Salmonella*-related investigations have shown prevalences varying from 2% to 20% [24]. According to the literature [20,21,22,23], the expected prevalence of *Salmonella* spp. in an EG cluster was set at 10%. The sample size was calculated following the formula below of infinite population correction (Table 2).
$$n^{\prime }=\frac{NZ^{2}P\left(P-1\right)}{d^{2}\left(N-1\right)+Z^{2}P(1-P)}$$

*N* = number of animals in the cluster.

*Z* = level of confidence = 95%.

*P* = expected proportion = 10%.

*D* = precision of proportion = 5%.

**Table 2 animals-13-03403-t002:** Calculated sample sizes and respective number of animals in a cluster.

Number of Animals per Cluster	Calculated Sample Size (Rounded to Number of Sampled Animals)
2	2
5	4
10	8
20	13
70	22

From 179 observed urban EG, 83 scat samples were collected from adults (*n* = 83) and 24 from pulli (*n* = 24) accompanied by their parents. Most of the sampled birds were living permanently in a synanthropic manner, but free-ranging animals were also sampled in peripheral parts of cities.

### 2.2. Boot Cover and Cecal Samples

For a better understanding of potential *Salmonella* transmission routes to humans via EG droppings, heavily contaminated areas with frequent EG–human interactions, such as urban and public swimming pools, were selected here (Figure 1b and Figure 2). In all swimming pools, boot cover sample collection was conducted as it is the preferred method in food safety control (FSC) of poultry industrial farms. These utensils are simple elastic cotton tubes which are pulled over either boots or shoes (Figure 1b) [37]. This method allows a large-scale sampling of an area and is comparable to or even more sensitive than the collection of individual scat samples [38].

Several German urban public swimming pools were identified with either a temporal or a permanent EG population. In total, 14 public pools allocated in the federal states of Hesse and Baden Wuerttemberg in six different towns were complaining of frequent EG dropping contamination in their areas. Three of them were public pools on the border of a natural freshwater body. Most of them had particular contamination problems around the heated baby/children swimming pools due to resting EG before and after the opening hours of the pool. These 14 public swimming pools were inspected to identify other waterfowl species. Most of them habituated not only EG but also Canada geese. At minimum two and at maximum six sock samples, depending on the size of the area and the number of geese, were taken. In total, 30 pairs of boot sock samples (*n* = 30) were taken. This sampling method was carried out following commission regulation (EU) No. 517/2011, paragraph 3.1.1. Plastic socks were put over shoes while sampling to prevent the contamination of different samples.

Taking into account that previous studies on *Salmonella-*inoculated chickens showed that not only the highest number of bacteria are found in caecum but that also that *Salmonella* spp. persist three times longer in this large intestine section [39], we decided to also include EG cecal samples of deceased animals. Twenty-five pairs of caeca from freshly dead EG, shot by hunters in Lower Saxony, were incubated in pools of 10 ceca.

### 2.3. Salmonella Detection

The scat samples were analysed following the *Salmonella* enrichment procedure described by the *Salmonella* reference laboratory in Bilthoven (ISO 6579 (2017)) [40]. This method is the European and international standard method for detection of *Salmonella* spp. [41].

In brief, a minimum of 10 g faeces were incubated in buffered peptone water (1:10) at 37 °C for 18–24 h. Thereafter, three drops of the surface of this pre-enrichment culture were given on the modified-half solid-Rappaport-Vassiliadis Medium (MSRV). The MRSV culture was incubated for 48 h at 41 °C. After 24 h of incubation, the surface of the medium was controlled for bacterial growth for the first time. Suspect swarming colonies were transferred on two selective isolation agars. These were xylose-lysin-desoxycholat-agar (XLD) and brilliant-green phenol-red lactose sucrose agar (BPLS), which were incubated at 37 °C for 24–48 h. *Salmonella* formed on XLD medium black colonies.

The elastic cotton tubes as well as the cecal samples were analysed with a combination of enrichment cultivation and a specific PCR. In order to record transmission pathways and the general contamination of swimming pool lawns, not only living germs should be recorded, but also non-infectious debris and viable but non-culturable (VBNC) germs. Therefore, a specific PCR was used for boot covers and cecal samples. The use of buffered peptone water as an enrichment cultivation increased the recovery of *Salmonella* in environmental samples by approximately 25% [42]. As already stated elsewhere, PCR methods are known to be as sensitive as the ISO 6579 (2017) for the detection of *Salmonella* spp. [43], and moreover they are able to detect non-swarming serovars of *Salmonella* spp. like *S.* Gallinarum nonetheless these serovars are in general of less zoonotic potential [44]. The samples were pre-incubated in buffered peptone water for 24 h at 37 °C to allow the best conditions for bacterial multiplication and to increase the sensitivity of specific PCRs. The elastic cotton tubes as well as the cecal sample pools were incubated in pairs in at minimum 225 mL of buffered peptone water for each pair. Further preparations and approaches were conducted following the instructions of the commercial qPCR-Kit *Salmonella* spp. from Kylt^®^. The kit is certified by the Friedrich-Löffler-Institute (FLI-B-656) according to § 11 paragraph 2 of the German Animal Health Act.

### 2.4. In Vitro Testing of Salmonella spp. Survival in Chlorinated Pool Water

In vitro tests of *Salmonella* spp. survival in EG droppings in tempered, chlorinated swimming pool water were implemented. Therefore, the intestinal content of EG was first frozen at −20 °C for several months to reduce the germ content. That content was divided in portions of 4 g and formed in artificial EG droppings. These ‘artificial EG droppings’ were then tested on XLD-agar for potential *Salmonella* spp. content.

*S.* Anatum (ATCC 9270) colonies grown on Columbia agar with 5% sheep blood were picked up and dispersed in 200 µL 0.9% NaCl solution. The number of cells was determined by triple dilution in a Neubauer counting chamber. With this method, it could be verified that the number of cells was proportional to the number of colony-forming units (CFU). Finally, 200 µL 0.9% NaCl solution with 40 CFU contained 3.3 × 10^11^ cells/mL and 20 CFU in 200 µL 0.9% NaCl solution contained 1.5 × 10^11^ cells/mL.

Each preformed ‘artificial EG dropping’ was spiked with 200 µL of the solution. They were suspended in 200 mL of four different swimming pool chlorine concentrations (please refer to Table 3). The fresh pool water sample containing chlorine concentrations demanded for public swimming pools was kindly provided by the “Sportschwimmhalle Dessau”, Dessau, Germany. Tap water (200 mL) was used as a negative control. Chlorine concentrations varying between 0.3 and 0.6 mg/L are the demanded ones by German authorities for all public pools according to the DIN 19643 (2023) norm [45].

The beakers were incubated at 25 °C to simulate a tempered swimming pool. After 30 min, 60 min and 6 h incubation, 200 µL of the solution was plated on XLD-agar plates.

These XLD-agar plates were incubated for 18 h at 35 °C and suspected colonies were identified by MALDI-TOF MS (Bruker, Bremen, Germany) analysis.

## 3. Results

Herein, we re-confirm the permanent presence of wild neozootic EG not only in urban parks, riversides and artificial urban ponds, but also in public swimming pools in various German cities. Most of the employees of public swimming pools were reporting either single EG pairs, particularly during spring and early summer, but also on increasingly notoriously greater numbers, i.e., EG clusters, during late summer and autumn. These synanthropic EG populations in public swimming pools often share these spaces with free-ranging Canada geese and less frequently with mallards.

Interestingly, in public swimming pools particularly flat and well-tempered chlorinated water bodies, such as the ones for babies and/or toddlers, are frequently preferred as resting areas of neozootic EG. Irrespective of these epidemiological criteria, none of analysed individual EG scat samples (*n* = 179), including ‘boot covered’ collected samples, were positive for *Salmonella*. All inhibition controls resulted PCR-positive in the HEX-curve therefore indicating that there were no PCR-inhibiting subjects in collected samples.

Regarding the question of whether *Salmonella* might be capable of surviving for a longer period of time (i.e., 30, 60, 180 min) in EG droppings exposed to chlorinated water, we performed in vitro trials with *S.* Anatum-spiked ‘artificial EG droppings’. These trials showed the survival of *S.* Anatum in spiked EG droppings for all tested chlorine concentrations and for up to 180 min (Figure 3). As such, the *Salmonella*-positive XLD agars had the typical pink indication colour with black colonies in the edge regions. As expected, all bacterial colonies were pure *S.* Anatum cultures. The number of CFUs on XLD agar plates were too many to be counted manually. Additionally, for final bacterial species identification, a MALDI-TOF MS analysis was conducted, thereby confirming the identification of *Salmonella*.

## 4. Discussion

Wild bird species which are closely associated with human activities or livestock are more likely than other species to show higher prevalences of intestinal anthropozoonotic serovars of *Salmonella* because of increased human-derived environmental contamination [13,15,23]. Epizootiological scenarios of transmission occur in urban environments contaminated with droppings of synanthropic EG. These complex species–species interactions will also allow potential spill-overs of zoonotic *Salmonella* [46,47,48]. In addition, neozootic species such as EG are natural reservoirs of infectious pathogens which might threaten domestic animal and human health [2].

Overall, the prevalence of *Salmonella* spp. in German urban EG populations seems to be extremely low, as none of the investigated scat samples resulted positive for this bacteria species. Even if a slightly higher prevalence might be estimated by considering a bias from the cultivation methods, the real prevalence is obviously very low. Consequently, *Salmonella* spp. prevalences in wild waterfowl of other European countries differ very strongly and range from 0% up to 20% in some limited areas [23,24,33]. The obtained 0% prevalences in the sampled EG droppings are in accordance with previous German studies focusing on *Salmonella* infections in closely related species, i.e., free-living Canada geese, grey lag geese and ducks, where no serotypes of *Salmonella* were detected as well [16,17,18]. Neither species–specific nor unspecific serovars of *Salmonella* spp. were found in all these former investigations. These results might initially suggest the inexistence of permanent circulating *Salmonella* infections within wild waterfowl populations in Germany, but more epidemiological monitoring studies with much larger sample numbers are urgently needed for final conclusions on the role of wild EG as reservoir hosts. Similarly to waterfowls, a rather low *Salmonella* prevalence was found in rodent populations inhabiting public places of London, and most of them corresponding with *S.* Typhimurium serovars [24].

Another aspect that may lead to an underestimation of the prevalence in this work is an intermitting excretion and thus a reduction in detectability. *Salmonella* is known to be a bacterium that is intermittently shed by avian hosts [49]. Sampling single free-ranging individuals over several days is almost impossible in free-ranging birds. Therefore, sock samples and cecal samples were used in addition for these studies. Since *Salmonella* is able to persist in the environment up to a month, as shown by Feare et al., sock samples were taken as very sensitive environmental sample collection [24,50].

As reported for other waterfowl species, neozootic EG seem to be less susceptible to zoonotic *Salmonella* serovars circulating in other endemic wildlife and/or domestic animals. This alien bird species seems to be extremely adaptable to different environments, food supplies and infectious pathogens [2,6,51]. As reported for other alien species worldwide, the EG must possess a strong innate and adaptive immune system to efficiently combat new pathogens found in newly conquered habitats. In line with this, physical stress resulting from constant exposure to different or even adverse climate conditions should be less pronounced compared to other birds, as reflected in their fast spread into non-endemic European countries [2,9,52,53].

As mentioned above, described *Salmonella* spp. prevalences of wild waterfowl in other European countries differ very strongly and range from 0% up to 20% in some limited areas. Elmberg et al. posit that in some areas, prevalences might be that high because of contamination due to human waste and domestic pet defecation [33]. These authors recommended offering sanitary facilities for humans and dogs in parks and better hygiene procedures to avoid environmental contamination and the spread of this bacteria [13,33]. Conversely, in our present study no such heavily human waste-contaminated public areas were detected. Nonetheless, in certain urban parks or public swimming pools of larger cities, i.e., Frankfurt am Main or Mannheim, synanthropic EG populations might become exposed to human, pet or even rodent faeces. A study in another European city (Barcelona, Spain) was able to detect *S.* Anatum and *S.* Corvallis in urban wild boars (*n* = 41, 2 positive). Navarro-Gonzalez et al. also point out the comparatively close direct and indirect contact with civilisation through waste consumption, urban water surfaces and excretions [54]. Moreover, spill-over and spill-back between farm animals and urban areas would be possible for all free-ranging animals moving between rural and urban habitats.

Most of the swimming pool employees confirmed that the preferred places of wild EG in public pool gardens are frequently found around the baby/toddler pools, where these animals like to stand in the flat waters and/or graze around these pools. Unfortunately, toddlers as well as children under five years are the most exposed group in the population for salmonellosis [28]. Marked geophagy and less hygiene awareness in this cohort clearly increase the risk of oral infection [30]. More importantly, children are more susceptible to developing severe gastroenteritis and progression of salmonellosis [55].

Our in vitro trial on chlorinated water showed very clearly the survival of *S.* Anatum in tempered swimming pool water, therefore justifying EG protection measures for public swimming pools as recently proposed by some German city municipalities [56]. Different former investigations tested much higher chlorine concentrations to disinfect different organic material such as carcasses or seeds contaminated with *Salmonella* spp. with more or less success [57,58]. All authors reported on the chlorine resistance of different *Salmonella* strains, thereby demanding future detailed investigations on disinfection strategies. Which organic material is disinfected by chlorination seems to be decisive, but in contrast the temperature seems to have no influence on *Salmonella* survival [57,59,60,61]. Organic materials with high protein contents are prone to promote bacterial survival, as especially amino acids (aa) are binding free chlorine molecules, thus reducing their ability to kill bacteria [58]. *Salmonella* spp. are excreted in geese droppings, which contain different concentrations of proteins depending on the nutrients and thereby building a ‘protective aa wall’ against free chlorine molecules [62]. Although the dilution of the droppings in the dimensions of a swimming pool is much higher than in the used small beakers of our in vitro trial, an infectivity over hours of the bacteria can be presumed.

Even though in this study no *Salmonella* spp. were detected in EG droppings, most probably due to the small sample size (*n* = 138), the risk of potential shedding of zoonotic serovars is given and an exclusion of this transmission risk is not possible. Therefore, yearly bacterial monitoring performed with boot covers of neozootic EG droppings, particularly in public swimming pools or urban ponds, should be recommended. The same holds true for scat samples originating from endemic waterfowl species visiting urban pools in order to avoid transmission to humans. Not only is the use of boot covers during scat sample collection important in order to increase sensitivity, but also the determination of *S. enterica* subsp. *enterica* serovars to evaluate the real zoonotic risk. Finally, high standard hygiene procedures are recommended for all employees of public pools as well as visitors of urban pools and parks, including strict hand hygiene and the avoidance of water drinking.

## 5. Conclusions

In contrast to the widespread acceptance that waterfowls like chickens are frequent spreaders and shedders of enteric *Salmonella* spp., this study did not find any positive synanthropic EG. Finally, to avoid the emergence of human salmonellosis in urban areas with abundant and permanent EG populations, the authors recommend the regular screening of predisposed sites with significant dropping contamination and additionally recommend high personal hygiene awareness.

## Figures and Tables

**Figure 1 animals-13-03403-f001:**
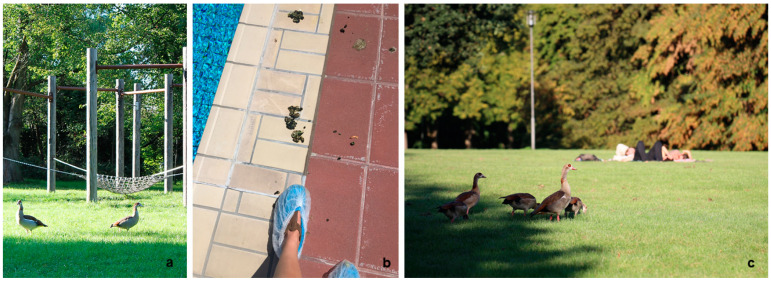
(**a**) Adult Egyptian geese (*Alopochen aegyptiaca*) in a public swimming pool park; (**b**) droppings of Egyptian geese around swimming pools and boot cover sampling procedure in urban swimming pools with a plastic boot cover and two cotton tubes; (**c**) Egyptian geese grazing on a lawn of a public swimming pool.

**Figure 2 animals-13-03403-f002:**
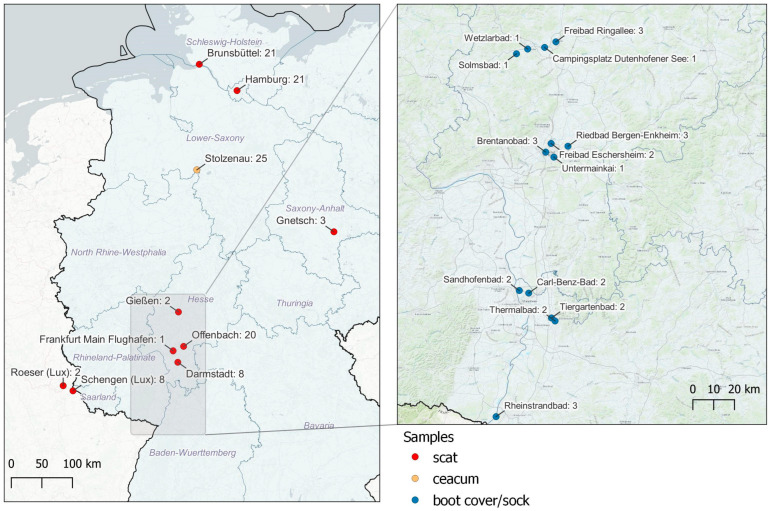
Sample sites in Germany and Luxembourg.

**Figure 3 animals-13-03403-f003:**
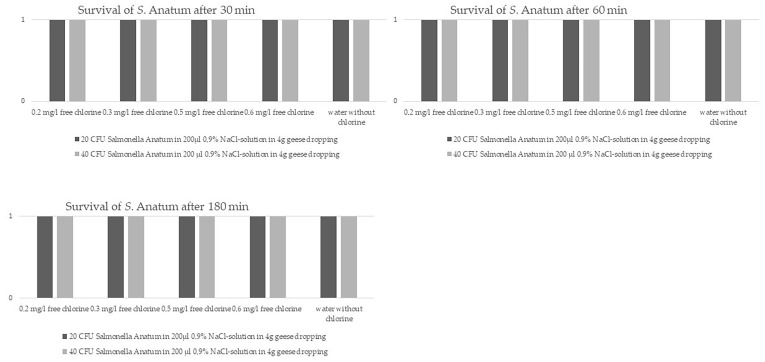
Survival of *Salmonella enterica* subsp. *entrica* serovar Anatum in different concentrations of chlorinated water after 30 min, 60 min and 180 min.

**Table 1 animals-13-03403-t001:** Detected prevalences of *Salmonella* spp. in waterfowl in Europe.

Host Species	Country/Locality	Number of Examinations (*n*)/ Number Infected	Reference
Canada geese (*Branta canadensis*)	Germany	289/0	Bönner et al. [16]
Grey lag geese (*Anser anser*)	Cologne Bay, Germany Niederrhein, Germany	175/0	Bolte et al. [17]
Brent Goose (*Branta bernicla*) Barnacle Goose (*Branta leucopsis*) Greylag Goose (*Anser anser*) White-fronted Goose (*Anser albifrons*) Pink-footed Goose (*Anser brachyrhynchus*) Bean Goose (*Anser fabalis*)	Sneek, The Netherlands Texel, The Netherlands Diepholz, Gemany Kampen/Zwolle, The Netherlands Wilhelmshafen, Germany Niederrhein, Germany	NG/0	Holländer [18]
Wild ducks (no further information given)	Bavaria, Germany	319/1	Thierfelder et al. [19]
Canada goose (*Branta canadensis*) Mallard (*Anas platyrhynchos*) Mandarin duck (*Aix galericulata)* Muscovy duck (*Cairina moschata)* Mute swan (*Cygnus olor*) Greylag goose (*Anser anser*)	Austria Czech Republic	50/0 51/2	Konicek et al. [20]
Mute swan (*Cygnus olor*)	Denmark	605/97	Nielsen et al. [21]
Greylag goose (*Anser anser*) Mallard (*Anas platyrhynchos*)	Rogaland country, Norway	219/1 5/0	Lillehaug et al. [22]
Eurasian teal (*Anas crecca)* Tufted duck (*Aythya fuligula)* Common pochard (*Aythya ferina)* Eurasian widgeon (*Mareca penelope)* Gadwall (*Mareca strepera)*	London, United Kingdom	80/3 198/10 130/4 12/0 15/0	Mitchell and Ridgwell [23]
Canada goose (*Branta canadensis*)	Yorkshire, United Kingdom London, United Kingdom	300/30 300/0	Feare et al. [24]

NG = not given.

**Table 3 animals-13-03403-t003:** Chlorine concentrations used to analyse survival of *Salmonella* Anatum in spiked EG droppings.

	20 CFU *Salmonella* Anatum in 200 µL 0.9% NaCl Solution in 4 g Geese Dropping	40 CFU *Salmonella* Anatum in 200 µL 0.9% NaCl Solution in 4 g Geese Dropping
0.2 mg/L free chlorine	200 µL on XLD agar	200 µL on XLD agar
0.3 mg/L free chlorine	200 µL on XLD agar	200 µL on XLD agar
0.5 mg/L free chlorine	200 µL on XLD agar	200 µL on XLD agar
0.6 mg/L free chlorine	200 µL on XLD agar	200 µL on XLD agar
water without chlorine	200 µL on XLD agar	200 µL on XLD agar

## Data Availability

Data are available on request only due to legal, or commercial reasons. Some public swimming-pool-operators wish to be treated anonymously.

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
