# Peer review of "Role of Free-Ranging Synanthropic Egyptian Geese (Alopochen aegyptiaca) as Natural Host Reservoirs for Salmonella spp. in Germany"

_animals, 2023, doi:10.3390/ani13213403_

Round 1
Reviewer 1 Report
Comments and Suggestions for Authors
I was honored to review the manuscript entitled “Role of free-ranging synanthropic Egyptian Geese (Alopochen aegyptiaca) as natural host reservoirs for Salmonella spp. in Germany” submitted to Animals.
Salmonella serotypes are zoonotic and can be transferred from animals and between humans. They usually invade only the gastrointestinal tract and cause salmonellosis, the symptoms of which can be resolved without antibiotics. The number of samples used in this study are low for studying about prevalence of this species. Herein, the reviewer thought that this manuscript needs to improve in structure. There are some points that the authors should correct them and still needs some change for improving the manuscript.
There are some points to correct:
1) Line 20, 23, 24, 28, 29, … Salmonella should be in italic form. Please check it throw the manuscript.
2) Line 25: Alopechen aegyptic should be italic.
3) Introduction is extremely long. Please provide a table for showing the prevalence of this bacteria in other regions.
4) Add information about the PCR method in Material and methods. Which primers did you use? Which type of PCR did you use? Which gene did you detect? Did you use the universal primer to detect bacteria in samples and send them for sequencing?
5) Provide PCR data as a supplementary material.
6) Results: There is irrelevant data in this section (line 270-279). You can use this data in discussion.
In conclusion I believe, in my opinion that the present manuscript can be accepted after major revision, for publication.
Author Response
Reviewer 1
Thank you very much for you time in reading our manuscript and for providing us very helpful advices. We have incorporated almost all of your suggestions.
Point 1: The introduction must be improved to provide sufficient background and include all relevant references.
Response: Thank you for your helpful advice. In order to shorten the introduction and to present the information from other publications more clearly, we have included a table. Please refer to table 1 in the revised manuscript.
Point 2: The methods must be improved.
Response: Thank you for your constructive advice. Following the advice of the second referee, the paragraph describing materials and methods has been reorganised into paragraphs 2.1 Scat samples, 2.2 Cecal samples and boot cover samples, 2.3 Salmonella detection and 2.4 in vitro testing of Salmonella. Please refer to the lines 137 – 256.
Point 3: The results must be improved to be clearly presented.
Response: Thank you for your attentive reading. Following the advice of the second reviewer, the results paragraph was revised. Following his suggestions, the paragraph was shortened to emphasise our own findings.
Point 4: Line 20, 23, 24, 28, 29, … Salmonella should be in italic form. Please check it throw the manuscript.
Response: Thank you for your attentive reading. We changed this in lines 9, 11, 17, 18, 22, 23, 24 and 29. Please refer to the revised manuscript.
Point 5: Line 25: Alopochen aegyptiaca should be italic.
Response: Thank you for your attentive reading. We changed this. Please refer to lines 25-26 in the revised manuscript.
Point 6: Introduction is extremely long. Please provide a table for showing the prevalence of this bacteria in other regions.
Response: Thank you again for this constructive comment. Like written above we deleted lines 83- 101 and created a table to present the given information better structured.
Point 7+8: Add information about the PCR method in Material and methods. Which primers did you use? Which type of PCR did you use? Which gene did you detect? Did you use the universal primer to detect bacteria in samples and send them for sequencing?
Response: Thank you for your attentive reading. Unfortunately, we used the commercially available PCR kit from Kylt® for our PCR experiments. We contacted Kylt® to get more information about the used Primers. But due to their trade secrecy, the company does not provide any information about the primers or the amplified gene. It is a real-time PCR, which we describe in the manuscript. Furthermore, we have added the approval number provided by the Friedrich-Löffler-Institut and the regulatory background in the manuscript. Please refer to lines 229- 231 in the revised manuscript.
Point 9: Results: There is irrelevant data in this section (line 270 - 279). You can use this data in discussion.
Response: Thank you for this observation. Since the second reviewer recommended to delete the paragraph about emerging infectious because in his opinion, salmonellosis cannot be considered an emerging disease. The latest EFSA reports, that the number of diagnosed cases of salmonellosis has been reduced. Please refer to lines 283- 291 in the revised manuscript.

Reviewer 2 Report
Comments and Suggestions for Authors
The manuscript submitted by Fischer et al. describes a Salmonella surveillance study in an alien species in Germany: the Egyptian goose. Although the results obtained regarding the presence of Salmonella are negative, I believe that their publication may be useful for future researchers. Furthermore, the experiment carried out to evaluate the survival of Salmonella in chlorinated water is very interesting and necessary.
However, before publication it needs an in-depth review:
Review the Salmonella italics in the abstract, since they are not there.
L23. Write salmonellosis in lowercase, please.
L45. Figure B does not fit what is described in these lines. Please review.
L68. Please, put the reference with the corresponding number.
L70-97. Summarize this information, highlighting the most relevant data, and reserve the rest for discussion.
L148-151. I believe that this information does not need to be here since it is understood that when a sample is sent to a reference laboratory it is to have more sensitive results.
L178-180. Move to L143 along with the map.
L183. Please change "examined by the..." to "analyzed following the..."
L183-193. This information should be in a different section. It should be clearer if you maintain 2.1 Scat and cecal samples; 2.2. Boot cover samples; 23. Salmonella detection; 2.4. In vitro testing of Salmonella survival.
L193. The image does not correspond to colonies compatible with Salmonella.
L194. When starting the sentence with a number, said number must be spelled out with letters. Please change it to "Twenty-five..."
FIGURE 1. Would it be possible to upload it to the introduction? I think it has much more potential, since the introduction talks about the problems of EG in parks, gardens, and swimming pools.
L210-217. It seems repetitive, please move the information to L139 and merge it with what is already in that section.
L218-227. I'm not understanding the structure very well... Were samples of natural habitats (section 2.1) collected on the one hand and and pools (section 2.2) on the other hand? If so, please change the titles of the sections, it isn't very clear since it seems that on the one hand scat and cecal samples are collected in all locations, and on the other hand the boot covers used for the collection of those samples.
L229-239. For the analysis of these samples, PCR was performed instead of culture to increase sensitivity. Why wasn't PCR done directly on all the samples?
L242-243. Be careful, freezing at -20ºC for several months can also affect the survival of Salmonella. Indicate how many months the samples were frozen, 1 month is not the same as 10...
L250. You should not start a sentence with a number, write the number in letters, or start with a connector such as "Then..."—the same on line 256.
L253: 200 ml or 200 ul?
L284. None of the analyzed individual EG scat samples...
Results, a figure showing the results obtained in the survival of Salmonella at different chlorine concentrations would be interesting.
L300-306. I completely agree with this paragraph, however, in my opinion, salmonellosis cannot be considered an emerging disease, since in poultry farming we have been fighting against it for a long time, and in wild birds, it has been detected for many years... In fact, in According to the latest EFSA reports, the number of diagnosed cases of salmonellosis has been reduced, partly thanks to the effectiveness of the control measures implemented more than 10 years ago.
L370. "Of paramount importance" sounds quite strange to me, please rephrase.
If you allow me a suggestion, it would be interesting in the discussion to mention two aspects of the research. The first is that through culture, the sensitivity is lower and therefore the prevalence may be slightly underestimated, despite the fact that Salmonella is a bacteria that grows very well under the conditions of the ISO standard. On the other hand, it is a bacteria with intermittent excretion, so to obtain a much more precise prevalence, the ideal would have been to collect fresh feces for three days in a row. However, logistically this type of sampling with wildlife is practically impossible. Even so, I consider it important to point this out for future studies that may be proposed when reviewing this study.
Moreover, I see a low number of references in the discussion, for example, in lines 341 to 347 not a single reference is mentioned and information is given about the age group with the greatest exposure and most susceptible to developing the disease. Another example is the information given in lines 358 to 361, there is no reference that confirms this information. Please review the discussion carefully so that it has greater dynamism and that everything is adequately referenced.
As for the conclusions, I believe that the limitations should not be here but in the discussion, and in the conclusions section only give the conclusions derived from your research.
Author Response
Thank you very much for you time in reading our manuscript and for providing us very helpful advices. We have incorporated all of your suggestions. Thank you very much for your well-structured advice, which helped a lot to improve the manuscript.
Point 1: The description of the methods must be improved.
Response: Thank you for this important advice. We rebuild the paragraph about the methods according to your suggestions in point 11.
Point 2: The conclusions must be improved.
Response: Thank you for your constructive advice. We deleted the limitations of our study in the paragraph “conclusion” and transferred them to the section “discussion”. The section “conclusions” presents only outcomes of our research.
Point 3: Review the Salmonella italics in the abstract, since they are not there.
Response: Thank you for your attentive reading. We changed this in the lines 9, 11, 17, 18, 22, 23, 24 and 29. Please refer to the revised manuscript.
Point 4: L23. Write salmonellosis in lowercase, please.
Response: Thank you for your attentive reading. We changed this misspelling. Please refer to line 23 in the revised manuscript.
Point 5: L45. Figure B does not fit what is described in these lines. Please review.
Response: Thank you for your attentive reading. We changed this and checked the references to the figures throughout the text. Please refer to line 44 in the revised manuscript.
Point 6: L68. Please, put the reference with the corresponding number.
Response: Thank you for your suggestion. We added the specific EU regulations in the reference list and give it a corresponding number in the text. Please refer to line 66 in the revised manuscript.
Point 7: L70-97. Summarize this information, highlighting the most relevant data, and reserve the rest for discussion.
Response: Thank you for this helpful advice. According to the suggestion of the first referee we added a table about the determined prevalences of other studies. In addition, we deleted the lines 83- 101. In this way the information is summarized and highlighted. Please refer to table 1 in the revised manuscript.
Point 8: L148-151. I believe that this information does not need to be here since it is understood that when a sample is sent to a reference laboratory it is to have more sensitive results.
Response: Thank you for your helpful advice. We deleted this information. Please refer to line 138-139 in the revised manuscript.
Point 9: L178-180. Move to L143 along with the map.
Response: Thank you for your constructive advice. We reordered material paragraph. Please refer to lines 140- 144 in the revised manuscript.
Point 10: L183. Please change "examined by the..." to "analysed following the..."
Response: Thank you for your attentive reading. We changed this. Please refer to line 203 in the revised manuscript.
Point 11: L183-193. This information should be in a different section. It should be clearer if you maintain 2.1 Scat and cecal samples; 2.2. Boot cover samples; 23. Salmonella detection; 2.4. In vitro testing of Salmonella survival.
Response: Thank you for your constructive advice. We rebuild the paragraph “methods” according to your suggestions and created the new section “2.3 Salmonella detection”. Please refer to lines 202-224 in the revised manuscript.
Point 12: L193. The image does not correspond to colonies compatible with Salmonella.
Response: Thank you for you attentive reading. We deleted this reference. Please refer to line 213 in the revised manuscript.
Point 13: L194. When starting the sentence with a number, said number must be spelled out with letters. Please change it to "Twenty-five..."
Response: Thank you for your attentive reading. We changed this. Please refer to line 199 in the revised manuscript.
Point 14: FIGURE 1. Would it be possible to upload it to the introduction? I think it has much more potential, since the introduction talks about the problems of EG in parks, gardens, and swimming pools.
Response: Thank you for this helpful advice. We moved up Fig. 1 up between the lines 67- 68 in the revised manuscript.
Point 15: L210-217. It seems repetitive, please move the information to L139 and merge it with what is already in that section.
Response: Thank you for your attentive reading. We deleted the repetitive phrases. Please refer to lines 175- 182 in the revised manuscript.
Point 16: L218-227. I'm not understanding the structure very well... Were samples of natural habitats (section 2.1) collected on the one hand and and pools (section 2.2) on the other hand? If so, please change the titles of the sections, it isn't very clear since it seems that on the one hand scat and cecal samples are collected in all locations, and on the other hand the boot covers used for the collection of those samples.
Response: Thank you for your kind words. We rebuild the paragraph “methods” by structuring it in four paragraphs and think it should be clearer now. The scat samples were examined individually. The sock swabs were examined in pairs and the cecal samples were examined in pools. Please refer to lines 136- 225 in the revised manuscript.
Point 17: L229-239. For the analysis of these samples, PCR was performed instead of culture to increase sensitivity. Why wasn't PCR done directly on all the samples?
Response: Thank you for this interesting question. With the collection of individual fecal samples, we wanted to determine a prevalence that could be assigned to individual animals. In this way infectious germs are detected, which might, depending on the serovar, infect humans. The sock swabs collected on the public pool lawns, were used to determine how high the contamination of areas shared by humans and Egyptian geese is. The use of buffered peptone water increases the recovery of Salmonella in environmental samples by approximately 25%. Since Salmonella remain infectious for up to one month in the free environment and the animals are not always present while sampling, this was a very sensitive method of choice to detected living germs as well as vbnc germs. We tried to make this clearer in lines 216- 221 in the revised manuscript.
Point 18: L242-243. Be careful, freezing at -20ºC for several months can also affect the survival of Salmonella. Indicate how many months the samples were frozen, 1 month is not the same as 10...
Response: Thank you for your attentive reading. Yes, we used freezing at -20 °C to be sure, that no possible Salmonella strains are culturable in the ingesta. To exclude further contamination the ingesta passed an enrichment in peptone and a following specific PCR. There after it was used for spiking experiment.
Point 19: L250. You should not start a sentence with a number, write the number in letters, or start with a connector such as "Then..."—the same on line 256.
Response: Thank you for your attentive reading. We changed these sentences. Please refer to lines 240 and 246 in the revised manuscript.
Point 20: L253: 200 ml or 200 ul?
Response: Thank you for your attentive reading. We checked this.
Point 21: L284. None of the analyzed individual EG scat samples...
Response: Thank you for your advice. We changed this. Please refer to line 267 in the revised manuscript.
Point 22: Results, a figure showing the results obtained in the survival of Salmonella at different chlorine concentrations would be interesting.
Response: Thank you for your constructive advice. We created a figure to show the result of the invitro trial. Please refer to Fig 2 in the revised manuscript.
Point 23: L300-306. I completely agree with this paragraph, however, in my opinion, salmonellosis cannot be considered an emerging disease, since in poultry farming we have been fighting against it for a long time, and in wild birds, it has been detected for many years... In fact, in According to the latest EFSA reports, the number of diagnosed cases of salmonellosis has been reduced, partly thanks to the effectiveness of the control measures implemented more than 10 years ago.
Response: Thank you for your attentive advice. We deleted the paragraph about emerging infectious diseases and restructured the paragraph “results”. Please refer to lines 283- 291 in the revised manuscript.
Point 24: L370. "Of paramount importance" sounds quite strange to me, please rephrase.
Response: Thank you for your attentive reading. We rephrased this sentence. Please refer to line 355 in the revised manuscript.
Point 25: If you allow me a suggestion, it would be interesting in the discussion to mention two aspects of the research. The first is that through culture, the sensitivity is lower and therefore the prevalence may be slightly underestimated, despite the fact that Salmonella is a bacteria that grows very well under the conditions of the ISO standard. On the other hand, it is a bacteria with intermittent excretion, so to obtain a much more precise prevalence, the ideal would have been to collect fresh feces for three days in a row. However, logistically this type of sampling with wildlife is practically impossible. Even so, I consider it important to point this out for future studies that may be proposed when reviewing this study.
Response: Thank you for your kind words and this helpful suggestion. We incorporated these points in our discussion. Please refer to lines 292- 295.
Point 26: Moreover, I see a low number of references in the discussion, for example, in lines 341 to 347 not a single reference is mentioned and information is given about the age group with the greatest exposure and most susceptible to developing the disease. Another example is the information given in lines 358 to 361, there is no reference that confirms this information. Please review the discussion carefully so that it has greater dynamism and that everything is adequately referenced.
Response: Thank you for your attentive reading. We added references in lines 297, 324, 319, 339, 340, 341, 347 and 355. Please refer to the revised manuscript.
Point 27: As for the conclusions, I believe that the limitations should not be here but in the discussion, and in the conclusions section only give the conclusions derived from your research.
Response: Thank you for your helpful advice. We deleted the limitations of our study in the paragraph “conclusion” and transferred them to the section “discussion”.

Reviewer 3 Report
Comments and Suggestions for Authors
Dear Ella and coauthors,
I have found your work very descriptive but timely and approapiate for the special issue. Even if your results are negative, provide very and opportune baseline data for salmonella monitoring in wildlife. However, I would greatly recommend putting your results in the context of other synurbic species such as the wild boar. In our studies, we have detected Salmonella prevalences of 5% (Navarro-Gonzalez, et al. Food-borne zoonotic pathogens and antimicrobial
resistance of indicator bacteria in urban wild boars in Barcelona. Spain. Vet.
Microbiol. 167, 686–689.) in boars coexisting with humans, visiting parks, playgrounds, and other urban areas. This Salmonella carriage in wild boar seems to be driven by livestock farming since cattle are the main Salmonella reservoir in the area. So, I am wondering whether Egyptian geese visit poultry, pig, or farms increasing the risk of infection spreading bacteria from farms to the urban environment. I think it would be interesting and adequate to introduce this idea in the discussion section.
Author Response
Thank you very much for you time in reading our manuscript and for providing us very helpful advices. We have incorporated your very interesting suggestion.
Point 1: I would greatly recommend putting your results in the context of other synurbic species such as the wild boar. In our studies, we have detected Salmonella prevalences of 5% (Navarro-Gonzalez, et al. Food-borne zoonotic pathogens and antimicrobial resistance of indicator bacteria in urban wild boars in Barcelona. Spain. Vet. Microbiol. 167, 686–689.) in boars coexisting with humans, visiting parks, playgrounds, and other urban areas. This Salmonella carriage in wild boar seems to be driven by livestock farming since cattle are the main Salmonella reservoir in the area. So, I am wondering whether Egyptian geese visit poultry, pig, or farms increasing the risk of infection spreading bacteria from farms to the urban environment. I think it would be interesting and adequate to introduce this idea in the discussion section.
Response: Thank you for this helpful advice. We added your paper to the references and discussed the direct and indirect contact between wildlife and humans in urban areas. Our field observations show, that urban Egygtian geese are used to human activities, often tame and more stationary than individuals in rural areas. The urban EGs are feeding on anthropogenic feeding and lawns in public parks and are maybe not forced to cover greater distances to farms or rural areas. But telemetric observations of their daily movement behaviour are unfortunately missing in Europe. We incorporated this idea in our manuscript. Please refer to line 334- 339 in the revised manuscript.

Round 2
Reviewer 1 Report
Comments and Suggestions for Authors
I would like to thank authors for preparing this version of the manuscript.
Here is some points:
Attention: all species should be italic in all sentences. check all pages and correct them in it.
1) Line 276: Salmonella should be italic.
2) Line 285: S. Anatum should be italic.
3) Line 254, 255: italic
4) Line 313: S. Typhimurium should be italic.
5)Line 341,354: italic